# Molecular and Phylogenetic Analyses of Lumpy Skin Disease Virus (LSDV) Outbreak (2021/22) in Pakistan Indicate Involvement of a Clade 1.2 LSDV Strain

**DOI:** 10.3390/v17121546

**Published:** 2025-11-26

**Authors:** Saiba Ferdoos, Andy Haegeman, Sadia Sattar, Ibrar Ahmed, Sundus Javed, Aamira Tariq, Nick De Regge, Nazish Bostan

**Affiliations:** 1Molecular Virology Labs, Department of Biosciences, COMSATS University Islamabad, Islamabad 45550, Pakistan; saibaferdoos@yahoo.com (S.F.); sadia.sattar@comsats.edu.pk (S.S.); 2Sciensano, Infectious Diseases in Animals, Exotic and Vector-Borne Diseases, 1180 Brussels, Belgium; andy.haegeman@sciensano.be; 3Alpha Genomics Private Limited, Islamabad 45710, Pakistan; alphagenomics.co@gmail.com; 4Microbiology and Public Health Labs, Comsats University Islamabad, Islamabad 45550, Pakistan; sundus.javed@comsats.edu.pk (S.J.);

**Keywords:** lumpy skin disease, Pakistan, phylogenetic analysis, *EEV*, *B22R*, *GPCR*, *P32*, *RPO30*

## Abstract

Livestock is the backbone of the economy in an agricultural country like Pakistan, with cattle serving as a milk and protein source. In 2021/22, Pakistan was hit by the first major outbreak of lumpy skin disease (LSD) in cattle, in all four provinces. LSD is characterized by the development of skin nodules, leading to severe illness, decreased milk production, and mortality, causing huge economic losses. This study aimed to analyze and classify the lumpy skin disease virus (LSDV) strains involved in the outbreak in the Punjab province at the molecular and phylogenetic levels to develop effective control strategies. A combination of different real-time PCRs was used for the identification and differentiation between vaccine, wild-type, and recombinant LSDV strains. This was mented with the sequence determination and phylogenetic analysis of ten genomic loci from two selected isolates from the 2021/22 Pakistan outbreak. The combined data showed that these isolates belonged to LSDV clade 1.2 and were clearly different from the vaccine clade 1.1 (Neethling-like), as well as from the recombinant clade 2 strains. In addition, using a fit-for-purpose gel-based PCR, the isolates from the outbreak were also shown to be different from KSGP0240-based vaccines.

## 1. Introduction

Pakistan is an agricultural country where a vast majority of the population relies on cattle and household animals for milk and meat supply [1]. The lumpy skin disease (LSD) outbreak in 2021–2023 has led to enormous economic losses in terms of reduced milk production, treatment of infected animals, abortions, and even death [2]. In Pakistan, LSD was reported for the first time in Jamshoro, Sindh, in November 2021. In the Sindh province, 20,000 animals were infected, and 54 deaths were recorded. Since its first reporting, the outbreak has widely spread across all four provinces, leading to extensive livestock damage. In Punjab, 35,000 LSD cases were reported, in Baluchistan, 22,225, in Khyber Pakhtunkhwa (KPK), 74,590 cases, and, in Azad Jammu and Kashmir (AJ and K), 6351 cases were documented. Nationwide, 190,000 cases of LSD were reported in cattle, with 7500 deaths. A proportion of animals recovered from the clinical disease [3,4]. It has put the country in a state of emergency for the control and prevention of LSD in all provinces. LSD is reported to infect cattle and water buffaloes; only a very low prevalence in water buffaloes (9.3%) was reported during the outbreak in Pakistan, potentially due to species differences [5]. An LSDV seroprevalence of 19.38% (*n* = 800) was reported in the Potohar region of Pakistan; out of these, 20.83% were cattle (*n* = 166) and 9.61% were water buffaloes (*n* = 77).

LSD is caused by the lumpy skin disease virus (LSDV), which belongs to the genus Capripoxvirus in the family *Poxviridae* with two other viruses, namely the goatpox virus (GTPV) and sheeppox virus (SPPV) [6]. LSDV has a 151 Kb double-stranded DNA genome that encodes 156 putative proteins [7]. The viral genome has 2.4 kb inverted terminal repeats on both ends [8]. An infection with LSDV is characterized by the appearance of skin nodules, fever, nasal discharge, and lymphadenopathy [9,10]. The severity of symptoms varies depending on the viral strain, breed, and age of the host, and its immune status [11]. LSD is a highly contagious viral disease, with a high morbidity and low to moderate mortality [12,13].

Since it was first reported in Zambia in 1929, the virus has spread across continents, reaching Southeast and South Asian countries [14,15,16]. The endemic LSDV strains in Africa and Southeast and South Asia can be divided into three main groups, named cluster 1.1, cluster 1.2, and cluster 2 [17]. The isolates of cluster 1.1 are the historical strains from South Africa, including the attenuated vaccine strain Neethling [18]. Cluster 1.2 contains isolates from Kenya, identified in the early 1950s, as well as the European and Middle Eastern isolates [19]. With the emergence of the new recombinant LSDV strains, a new distinct cluster 2 arose [20]. This cluster 2 can be further subdivided into six subclades, namely 2.1 to 2.6 [21]. While strains belonging to cluster 2.5 have spread across Asia [20,22,23,24], strains of the other five clusters remained limited to Russia [25,26,27]. These cluster 2 isolates from Russia and Southeast Asia showed more significant genomic variability compared to the historical strains [21,28,29,30]. These clade 2 strains are often referred to as recombinant LSDV strains. They show vaccine-like genomic profiles that are attributable to the recombination between the Neethling vaccine and the KSGP vaccine strain [31]. They are the result of a recombination that took place during an inappropriate vaccine production process containing both vaccine strains [30,31,32]. The emergence and spread of recombinant strains of LSDV in recent years make the constant monitoring and characterization of LSDV field isolates highly important [33].

The genetic and phylogenetic characterization of strains from an outbreak can provide valuable information about the origin of the disease, hotspot areas, and the level of transboundary circulation. It can also be helpful in developing eradication and control strategies [34,35]. Molecular characterization has for a long time relied on the analysis of various genes, such as *B22R* (a hallmark for differentiation between vaccine and wild-type strains), *GPCR* (a membrane-localized immunomodulatory protein), *RPO30* (the 30 KDa subunit of LSDV RNA polymerase), *P32* (an immunogenic surface protein similar to P35 of vaccinia virus), and the EEV surface glycoprotein genes (immunogenic target for vaccine development) [30,36,37,38]. In recent years, however, the whole-genome sequences of LSDV strains have been determined due to advances in sequencing technology.

This study was designed to characterize the LSDV outbreak strains from Pakistan. In the absence of a facility in which the virus could be isolated from field-collected samples, we had to work with samples with a limited viral load to try to determine the sequence. Therefore, rather than whole-genome sequencing, 10 genes were sequenced to ensure a correct phylogenetic placement.

## 2. Materials and Methods

### 2.1. Ethical Approval and Study Setting

The ethical approval (No. CUI-Reg/Notif-3089/22/3182) was obtained from the Ethical Review Board (ERB), COMSATS University, Islamabad. The study was non-intrusive, as all the samples were obtained from skin lesions by swabbing. The study area included different cities of the province of Punjab (*n* = 34), Khyber Pakhtunkhwa (KPK, *n* = 20), Sindh (*n* = 13), Azad Jammu and Kashmir (AJ&K, *n*= 7), and the Islamabad Capital Territory (*n*= 2) (Figure 1). Animals selected for sampling presented characteristic necrotic skin nodules.

### 2.2. Sample Collection and Nucleic Acid Extraction

Seventy samples were collected from selected areas of three provinces (Punjab, Sindh, Khyber Pakhtunkhwa) and the Islamabad Capital Territory (ICT) between March and November 2022. Out of these, fifty-nine were swab samples collected from infected animals that presented characteristic skin nodule lesions. The samples were taken from necrotic nodules through a sterilized swab stick contained in a sterile tube carrying 3 mL of Phosphate-Buffered Saline (PBS). In addition, 15 blood samples and 2 scabs were also collected. The samples were transported to the Molecular Virology Laboratory (MVL) using standard procedures and stored at 4 °C until further processing. DNA was extracted using protocols described previously [39], using Macherey-Nagel^TM^ NucleoSpin^TM^ Blood kit (Düren, Germany) following the manufacturer’s instructions with a few modifications. Lysis was carried out for an extended period of 1 h, and an external control was added to the B3 buffer before extraction. This section of work was performed in collaboration with the LSDV reference lab at Sciensano, Infectious Diseases in Animals, Exotic and Vector-Borne Diseases, Groeselenberg, Brussels, Belgium.

### 2.3. Application of Real-Time PCR Assays for Genus, Species, and Clade Identifications

Presence of LSDV DNA was confirmed by subjecting samples to a panCapripox real-time PCR using methods described by Haegeman et al. in 2013 [39]. The Wolff assay described by Wolff et al. in 2021 [40] was used to differentiate between different species of the Capripox genus, being LSDV, goatpox, and sheeppox virus. Further characterization of confirmed LSDV strains was carried out using DIVA PCRs as described by Haegeman et al. in 2023 [41]. This real-time PCR differentiates clade 1.1 Neethling vaccine strains from clade 1.2 and clade 2.5 strains.

### 2.4. Molecular Detection and Sequence Analysis of Ten Genomic Regions

#### 2.4.1. Amplification of Selected Regions for Sequence Analysis

Since the majority of reports from Pakistan rely only on sequences of one or two genes for phylogeny, we selected 10 genomic regions to obtain a comprehensive phylogenetic placement. The selected genomic regions, primer sequences, and annealing temperatures are listed in Table 1. The PCR mixes and cycling profiles were carried out as described in the corresponding publications given as references. Briefly, the PCR was carried out by mixing 2 μL DNA, 5 μL of Taq 10× buffer, MgCl2 (2.5 mM), dNTPs (Roche Applied Science, Baden-Württemberg, Germany), 2 units of Taq DNA polymerase (Life Technologies, Carlsbad, CA, USA), and 1 µL of each primer. The following cycling profile was used: 95 °C for 5 min (initial denaturation), followed by 40 cycles of 95 °C for 30 s; appropriate annealing temperature for each primer for 30 s; amplification at 72 °C for 30 s; and final amplification at 72 °C for 10 min.

#### 2.4.2. Differentiation Between KSGP0240-like Vaccine Strain and EUROPEAN/Middle Eastern LSDV Strains

New primers were designed upon an alignment by MAFFT [46] of the following publicly available genotype 1.2 sequences: Evros/GR/15 (KY829023), KSGP0240 (KX683219), Kenya (MN072619), and 155920/2012 (KX894508). A section of the inverted terminal repeat region was targeted, displaying a significant length difference between KSGP0240/Kenya and Evros/155920/2012. Using Primer3 software version 4.1.0 [47] (source code available at https://primer3.ut.ee/; accessed on 10 April 2025, a forward (GGTGAAATATTTTGAAGCCAAT) and a reverse primer (TTCGAGACCTCGTTTCTGAC) were obtained, amplifying an amplicon of either 278 bp (Evros)/263 bp (155920/2012) or 376 bp (KSGP0240). The 50 μL PCR mix consisted of 2 μL Capx DNA template, 5 μL PCR Taq buffer, 2.5 mM MgCl_2_, 0.3 mM of each dNTP (Roche Applied Science, Germany), and 33.75 pmol of the forward and reverse primer and 1.25 U Taq Polymerase (Thermofisher Scientific, Waltham, MA, USA). The cycling profile used was 95 °C for 4 min, 35 cycles of 95 °C for 30 s, 55 °C for 30 s, and 72 °C for 60 s, followed by one cycle of 72 °C for 10 min.

#### 2.4.3. Gel Electrophoresis, Extraction, and Sequencing

Following amplification, the PCR products were loaded on a 1.5% agarose gel and verified under UV using GelRed from Biotium (Fremont, CA 94538, USA, cat. number 41003). The amplicons were excised and purified using a gel extraction kit (QIAquick Gel Extraction Kit; Qiagen, Nordrhein-Westfalen, Germany, Cat no./ID. 28704) as described by the manufacturer. Before submission for Sanger sequencing, the quantity and purity of the extraction products were verified using Nanodrop (Thermo Fisher Scientifics, Bremen, Germany) (ratio 260/280). The sequenced fragments from both samples 1.4 *(LSDV_1_NS_2022_PAK)* and 3.29 *(LSDV_2_NS_2022_PAK)* were submitted to GenBank under the following accession numbers: *B22R*; PV492546, PV492556, *EEV* gene; PV492547, PV492557, *P32*; PV492548, PV492558, *RPO30*; PV492549 (not sequenced from *LSDV_2_NS_2022_PAK*), *GPCR*; PV492550, PV492559, *NTPase*; PV492551, PV492560, *RNA polymerase 132 (RPO132)*; PV492552, PV492561, *VLTF-1*; PV492553, PV492562, *Finger Protein*; PV492554, PV492563, *Ser/Thr kinase*; PV492555 and PV492564 for samples *LSDV_1_NS_2022_PAK* and *LSDV_2_NS_2022_PAK*, respectively.

#### 2.4.4. Phylogenetic Analysis

The obtained sequences were manually assembled in GeneDoc (version 2.7, freely available at https://genedoc.software.informer.com/2.7/, accessed on 10 April 2025) and checked using BLASTn, version 2.16.0 [48]. Publicly available LSDV sequences (representative of Asia, Europe, and Africa, and including all known genotypes), as well as those from goatpox and sheeppox viruses for all 10 regions, were collected from GenBank (https://www.ncbi.nlm.nih.gov/genbank/, accessed on 4 November 2024) and aligned using CLUSTALW in MEGA version 11 [49]. Subsequently, a nucleotide substitution model was selected for each dataset based upon the Bayesian Information Criterion (BIC) and Akaike Information Criterion (AIC) scores. Evolutionary history was inferred using the neighbor-joining method and the maximum likelihood method in MEGA version 11. The percentage of replicate trees in which the associated taxa clustered together in the bootstrap test (1000 replicates) is shown next to the branches [50].

## 3. Results

### 3.1. Sample Quality Control and Genus Confirmation

Approximately 57% (23/40) of the collected swab samples tested positive for LSDV using the panCapripox real-time PCR (Table 2). The lowest percentage of positive swabs was collected in the KPK region (35.7%), but many of these swabs also tested negative for the internal control (IC), indicating a low sample quality. In general, the Cp values (crossing point where the sample fluorescence signal rises above the background level) for the virus (D5r) were relatively high, considering that the swabs were taken from lesions. The average Cp value of the positive samples was 35.8, with a standard deviation of 4. The average Cp value of the internal control was 38.67, with a standard deviation of 4.4. The importance of good sample quality and storage was confirmed by the fact that 91.3% of the Capripox-positive samples were also positive for the IC (21 out of 23), while only 14% of Capripox-positives were detected in the IC-negative samples (2 out of 14).

### 3.2. Sequence Analysis of 10 Genomic Regions

Based on the sampling area, the obtained Cp value, and the amount of material available, two samples were selected for further characterization: one from Punjab (Sample ID: *LSDV_1_NS_2022_PAK*) and another from the Kyber Pakhtunkhwa (KPK) province (Sample ID: *LSDV_2_NS_2022_PAK*). In the initial analysis, the samples were tested with the DIVA PCR to determine LSDV clade identification. This DIVA PCR showed that both samples, *LSDV_1_NS_2022_PAK* and *LSDV_2_NS_2022_PAK*, contained wild-type-like strains, and not the clade 1.1 Neethling vaccine strain (Table 3). To provide additional sequence information, ten viral genomic regions were PCR-amplified (Appendix A), purified, and Sanger-sequenced from both samples (Table 1), except for the *RPO30* region, which was sequenced from sample *LSDV_1_NS_2022_PAK* only, as *LSDV_2_NS_2022_PAK* had insufficient material.

The combined sequence length of all analyzed regions for sample *LSDV_1_NS_2022_PAK* was 7965 nucleotides, while this was 6827 nucleotides for sample *LSDV_2_NS_2022_PAK* (Table 4). Except for the obtained length difference, the sequences of both samples were 100% identical. The difference in the obtained length was linked to the quality of base calling at the ends of the fragments. The nucleotide differences found between the samples and the representative sequences for the different LSDV genotypes are represented in Table 4.

The nucleotide comparison between the samples from Pakistan and the representative database sequences clearly indicated the presence of lumpy skin disease virus that was closest to genotype 1.2 LSDV strains in the database. However, a difference was obtained within the LSDV genotype 1.2 between the European sequences from Serbia and Greece and the ones from Africa (Kenya and KSGP0240). All the new recombinant strains (genotype 2.1 to 2.6) had at least 81 nucleotide differences from the Pakistan samples. The latter also had 120 or more nucleotide differences from the three representatives of the vaccine genotype 1.1. The sequence analysis of these regions was in full agreement with the initial DIVA PCR results.

### 3.3. Differentiation Between Genotype 1.2 Strains by Gel-Based PCR

As LSDV clade 1.2 also contains the KSGP0240 vaccine strain, a fit-for-purpose, gel-based PCR was developed to differentiate this vaccine (or vaccine-like strains) from the other clade 1.2 strains. Primers were designed to target a part of the inverted terminal repeat region in the LSDV genome displaying a 96 bp length difference between the European/Middle Eastern and African isolates using several reference sequences. This difference was confirmed by testing clade 1.2 strains/isolates, including Evros (KY829023), 155920/2012 (KX894508), and the vaccine strain KSGP0240 (KX683219).

The amplicons of seven samples from the Pakistan samples, including *LSDV_1_NS_2022_PAK* and *LSDV_2_NS_2022_PAK*, migrated at a similar length as those from Evros (KY829023) and 155920/2012 (KX894508) (Appendix A), and thus different from KSGP0240 (KX683219). The identity of the fragments and the sequence length difference were confirmed by Sanger sequencing of the amplicons from a sample from Pakistan (sample ID 2.20) and from KSGP0240. The PCR and sequencing results fully supported the previous sequence analysis of the *GPCR* region, which showed a closer match between samples from Pakistan and the European/Middle Eastern sequences.

### 3.4. Amino Acid Sequence Comparison of Amplified Regions

To understand the impact of the nucleotide changes on the protein level, the amino acid sequences of all 10 regions were compared to the corresponding regions of closely related LSDV strains in phylogeny (Table 1, Figure 2A–C, Appendix A). Out of these 10 genes, no amino acid changes were observed in *NTPase* and *VLTF-1*. *RPO132* and *B22R* were not the complete gene sequences and hence were not included.

The GPCR amino acid sequence (1098 bp, 366 AA; accession No. PV492559.1) exhibited a four-amino-acid deletion (TILS). These four amino acids—threonine-30, isoleucine-31, leucine-32, and serine-33—are present in the strains *LSDV/China/SX/2023* (PP894832) and *Vietnam 20L43_Ly-Quoc/VNM/20* (MZ577074); however, they are missing in isolates from Israel (*155920/2012*, KX894508), Pakistan *LSDV_2_NS_2022_PAK CC* (PV492559), and Albania *LSDV/Albania/4192/2016* (OR134835). A similar deletion has also been reported in the GPCR amino acid sequence of an Indian isolate *LSDV/02/KASH/IND/2022* (OQ588787) (Figure 2A). Additionally, a threonine at position 49 is missing in the GPCR protein of the isolate from Pakistan, whereas it is present in all other strains, including the Indian isolate.

The **EEV** glycoprotein from an isolate of Pakistan *LSDV_2_NS_2022_PAK* (PV492557) has an amino acid substitution E85K. A similar substitution has also been reported in an isolate from Bangladesh. In contrast, the Kenyan isolate *Kubash/KAZ/16* (MN642592), the Albanian isolate *LSDV/Albania/4770/2016* (OR134836), and the Neethling strain (AF409138) have glutamic acid at position 85, whereas the Indian isolate *LSDV/2022/Jamnagar/N1* (OR393167) and the isolate from Pakistan *LSDV_2_NS_2022_PAK* (PV492557) have lysine. These two amino acids have opposite charges.

Another amino acid substitution, I168M, is present in the isolates from Pakistan (*LSDV_2_NS_2022_PAK*, PV492557), India (*LSDV/2022/Jamnagar/N1*, OR393167), and Kenya (*Kubash/KAZ/16*, MN642592). The Albanian strains retain isoleucine at this position, while strains from Pakistan, Kenya, India, and Neethling all have methionine at position 168. Both amino acids (isoleucine and methionine) are hydrophobic and neutral in charge (Figure 2B).

In the RPO30 protein of *LSDV_1_NS_2022_PAK*, an amino acid substitution at position 14 (T14N) was observed. Threonine, in other closely related *RPO30* genes in phylogeny, was substituted by asparagine in *LSDV_1_NS_2022_PAK*. Both amino acids are hydrophilic and polar in nature (Figure 2C).

The P32 envelop protein gene of LSDV is widely used as a genetic marker because it is conserved in the genus. The P32 protein of the isolate from Pakistan *LSDV_1_NS_2022_PAK* was 100% identical to isolates from India *LSDV/Cattle/India/2019/Ranchi-1/P50* (OK422494), Serbia ERBIA/Bujanovac/2016 (KY702007), and Albania (*SDV/Albania/1707/2016*, OR134834). However, an isolate from Russia *LSDV/Russia/Udmurtiya/2019* (MT134042) has a substitution of valine to isoleucine at position 272 (**V272I**). Both amino acids are of similar polarity (Appendix A).

The fusion protein encoded by ORF117 is responsible for the fusion of the viral envelope with the host membrane. The protein sequence of the isolate from Pakistan *LSDV_1_NS_2022_PAK* (PV492552) was identical to isolates from India *LSDV/2022/Nohar* (OR393178) and Albania (*LSDV/Albania/790/2017*, OR134837). However, another isolate from India (*ICAR/NIVEDI/LSDV/Gaur/Karntaka/2023/India*, PQ510117) and an isolate from China (*LSDV/Jiling/2022*, OR567413) have a tyrosin substituted for a histidin at position 30 (**Y30H**). Both amino acids are polar (Appendix A).

### 3.5. Phylogenetic Analysis

A phylogenetic evaluation was carried out for all ten regions for both isolates from Pakistan (LSDV_1_NS_2022_PAK and LSDV_2_NS_2022_PAK), using both neighbor-joining (NJ) and maximum likelihood (ML) analyses. The most suitable nucleotide substitution models were selected for each dataset and are listed in Appendix A, along with other implemented parameters.

While the isolates from Pakistan clustered with different LSDV genotypes depending on the genomic region analyzed, the clade 1.2 sequences were the only ones that consistently grouped with them regardless of the method used (neighbor-joining or maximum liklihood). Examples of the evolutionary inference are depicted in Figure 3A–E. Overall, when all regions were considered together, the phylogenetic mapping indicated that the LSDV strains responsible for the 2021–2022 outbreak in Pakistan belonged to clade 1.2.

However, when we examined the individual trees, different clustering patterns were observed depending upon the genomic region analyzed. For example, the *B22R* region of *LSDV_1_NS_2022_PAK* (Figure 3A) clustered closely with the Rusian isolate *LSDV/Russia/Tyumen/2019* (OL542833) and the Albanian strain *LSDV/Albania/4192/2016* (OR134835). In contrast, the *EEV* gene (Figure 3B) exhibited the closest relationship with the Jmnagarh strain from India (LSDV/2022/Jamnagar/N1, OR393167), making a sister clade. Similarly, *GPCR* and *P32* map closely with strains from Russia *LSDV/Russia/Udmurtiya/2019* (MT134042) and strains from Serbia *SERBIA/Bujanovac/2016* (KY702007) (Figure 3C,D).

## 4. Discussion

LSD is an economically important and transboundary viral animal disease recognized by WOAH (World Organization for Animal Health) [51]. It has been one of the most devastating and emerging threats to domestic animals, wild bovines, and water buffaloes in recent years [11,52]. The current study aimed to characterize the LSDV strains from the first large-scale outbreak in the Punjab and Khyber Pakhtunkhwa provinces of Pakistan in 2021–2022 through PCR, nucleotide, and phylogenetic analyses of ten targeted genes: *RPO30* (ORF36), *GPCR* (ORF11), *B22R* (ORF134), *P32* (ORF74), *EEV* (ORF126), *NTPase* (ORF83), *RNA polymerase subunit 132* (ORF116), *Late Transcription Factor VLTF-1* (ORF58), *Finger Protein* (ORF10), and *Ser/Thr protein kinase* (ORF25) [28]. The presence of LSDV was already confirmed in the geographical region surrounding Pakistan, including India, Bangladesh, UAE, and China [51,53,54,55,56].

During this study, samples were collected by swabbing the lesions/nodules of clinically suspected animals. The low positivity rate of these swabs and the high Cp values in the positive samples show that external swabbing may not be the most suitable method for sample collection when the highest sensitivity is required (first case detection, import/export, etc.). Nevertheless, it remains an interesting way of taking additional samples, as it is easy and simple to collect in situations where the individual status is of less importance and an answer on the herd level is sufficient. When taking such a type of swab, it is important to verify that they are collected correctly, which can be monitored by looking for the presence of host skin material (hair, scabs, and blood traces). Alternatively, the use of an internal control to detect host material is a useful tool to avoid potential false-negative results. In this study, positive swabs were almost exclusively obtained when the internal control showed positive results. Notwithstanding this issue, LSDV was confirmed in cattle from Pakistan using the panCapripox real-time PCR targeting the D5r region. Using a DIVA real-time PCR, designed by Andy Haegeman in 2023, to differentiate between homologous vaccine strains (clade 1.1) and classical (clade 1.2) and recombinant (clade 2.5) LSDV strains, the isolates were shown to be distinct from the vaccine clade 1.1, thereby successfully validating this assay in our study [41].

As mentioned earlier, the LSDV strains can be divided into three distinct clusters: 1.1, 1.2, and cluster 2. The different genomic regions of both isolates from Pakistan clustered with different isolates from Bengal, India, China, Russia, Albania, Serbia, and Tibet. Most importantly, the isolates from Pakistan always clustered with clade 1.2 strains and never with clade 1.1 (vaccine) strains, independently of the region used. However, when looking at the different regions, LSDV strains belonging to one or more of the recombinant clades (2.1 to 2.6) also sometimes clustered with the isolates from Pakistan, but which recombinant clade this was depended on the specific region analyzed. It is only when the information of all the trees is put together that it can be concluded that the isolates from Pakistan belong to clade 1.2. This is consistent with a recent report from UVAS Pakistan that presented similar results based on whole-genome and individual gene sequences (OQ566164, OQ566165, OQ589501, OQ589502, OP807845-OP807849) [3,57]. The *GPCR* gene of isolates from Pakistan was grouped with those from Israel (KX894508), Kenya (MN072619), and India (OR393173), along with several others from the same region. The *P32* gene clustered with those from India (OR393176), Albania (OR134835), and Russia (MT134042). The *RPO30* gene clustered with West Bengal (OQ427097), Serbian (OR134847), Russian (MN995838), and Albanian isolates (OR134837). The *B22R* gene was closely related to isolates from West Bengal (OQ427097), India Jamnagar (OR393167), and China Tibet (OR797612). When we compared the *EEV* gene from both isolates, it was also closest to the India Jamnagar (OR393167) isolate, making a separate sister clade with it, in addition to KX894508 from Israel and the Kubash Kenya isolate (MN642592).

In this study, we compared the nucleotide sequences of selected genomic regions with their closest homologs in phylogeny. Six of the ten regions had no nucleotide differences between the isolates from Pakistan and genotype 1.2 sequences from Serbia and Greece (Evros GR/15). For the four other regions, only very limited differences could be observed (between one and three nucleotides). The nucleotide changes in the *EEV* fragment, encoding parts of LD126 (*EEV*) and LD127 proteins, of the samples from Pakistan resulted in only one amino acid change, namely lysine to glutamic acid. This is an interesting change, as lysine is positively charged, whereas glutamic acid is negatively charged. In the sequenced genome region encoding parts of a LAP/PHD finger-like protein (ORF10), the nucleotide change resulted in a positively charged lysine (Evros GR/15) becoming a hydrophobic non-charged phenylalanine. In the RNA polymerase subunit 30 kDa fragment, the nucleotide changes lead to a more conserved amino acid change, namely a threonine (Evros) to an asparagine (Pakistan), as both are non-polar and hydrophilic. In serine/threonine kinase (ORF25), a poly “T” section is extended in the isolates from Pakistan, leading to nine instead of seven “T’s”. This insertion leads to an altered 3′ end of the hypothetical protein LD026. Such an extension is similarly observed in isolates from India, Bangladesh, and China/Tibet between 2022 and 2024 (such as PQ472735, OR393178, and PP053747). Another interesting finding was the 12 bp difference in the *GPCR* gene between KSGP024/Kenya and the isolates from Pakistan. An identical difference in this gene was also observed in the European/Middle Eastern 1.2 strains. This finding indicates that the sequences of isolates from Pakistan are unrelated to the KSGP0240 vaccine, which was further confirmed through PCR, targeting a region in the inverted repeat region.

When looking at the different regions used for phylogenetic analysis individually, the resolution provided is very variable and not always able to solve the fine structure of the tree. This finding is consistent with the findings of Breman et al. [21]. Although the samples from Pakistan always clustered together with clade 1.2 isolates, they sometimes clustered with other genotypes. This poses an inherent risk of misclassification if only one region is included in a study. This can be circumvented by using multiple regions, as performed in this study, or by whole-genome sequencing (WGS). The latter was not attempted in this study because of the low viral load of the samples and the lack of sufficient quantity. Although some progress has been made in this regard, it remains a drawback of WGS.

The LSDV *EEV* glycoprotein gene is one of the most reliable and popular genetic markers used to distinguish LSDV field strains from vaccine strains in several studies [43] because it is based on the deletion of 27 bp in the vaccine strains. The multiple sequence alignments of the *EEV* gene in the current study showed the absence of the 27-nucleotide deletion in isolates from Pakistan, which differentiates between field isolates, Neethling-like vaccine strains, and LSDV recombinants from Southeast Asia and Russia.

## 5. Conclusions

This study was designed to evaluate the molecular and phylogenetic position of the LSDV present in samples collected during the 2021-22 LSDV outbreak in Pakistan. The application of DIVA PCRs showed that the isolates from Pakistan did not belong to the vaccine clade 1.1 and were not linked to the KSGP0240 vaccine, belonging to clade 1.2. Further sequence and phylogenetic analyses revealed that both isolates studied, *LSDV_1_NS_2022_PAK* and *LSDV_2_NS_2022_PAK*, belonged to clade 1.2 and were identical. It would be desirable to apply these strategies in large-scale studies for source tracking and the future prevention of outbreaks.

## Figures and Tables

**Figure 1 viruses-17-01546-f001:**
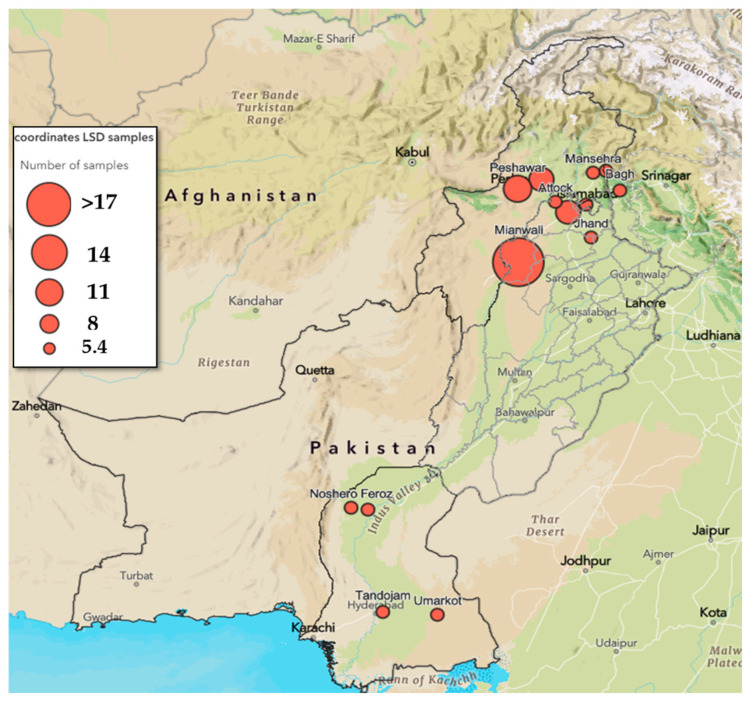
Sample collection sites shown on a map of Pakistan. Red dots represent sites, and the size of the dot indicates the sample number. Sample numbers are also provided in legends.

**Figure 2 viruses-17-01546-f002:**
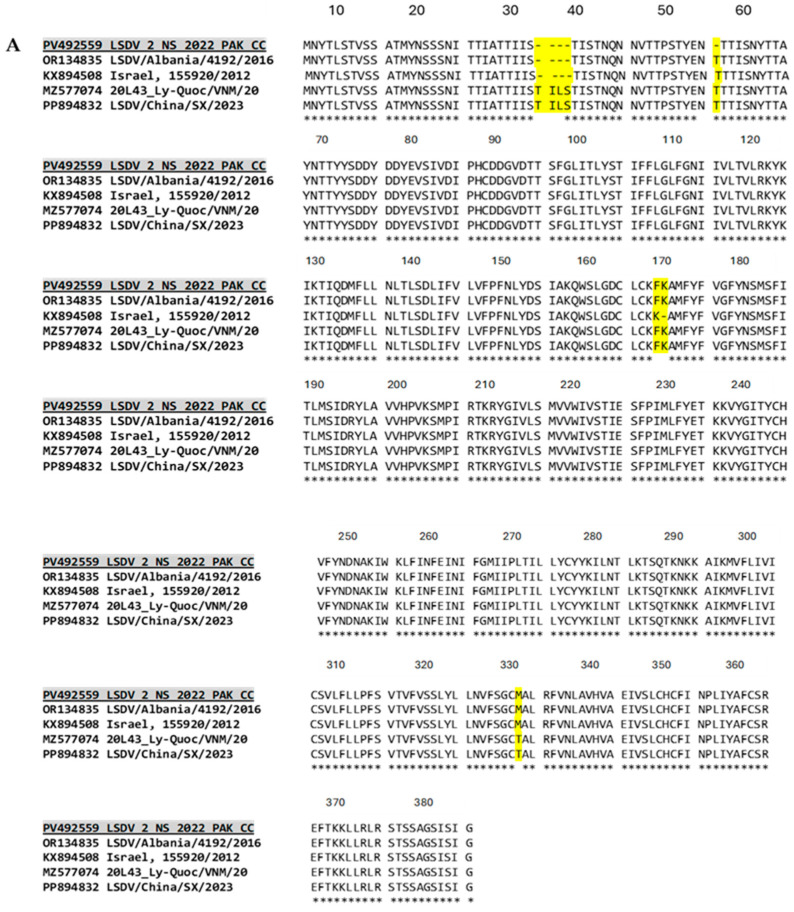
Amino acid sequence alignment of the *GPCR* gene (**A**) *EEV* gene (**B**) and *RPO30* gene (**C**) with their close relatives in phylogeny. The amino acid changes are highlighted in yellow. The accession number of the strain from Pakistan is highlighted in gray. The amino acid position is marked by numbers at the top.

**Figure 3 viruses-17-01546-f003:**
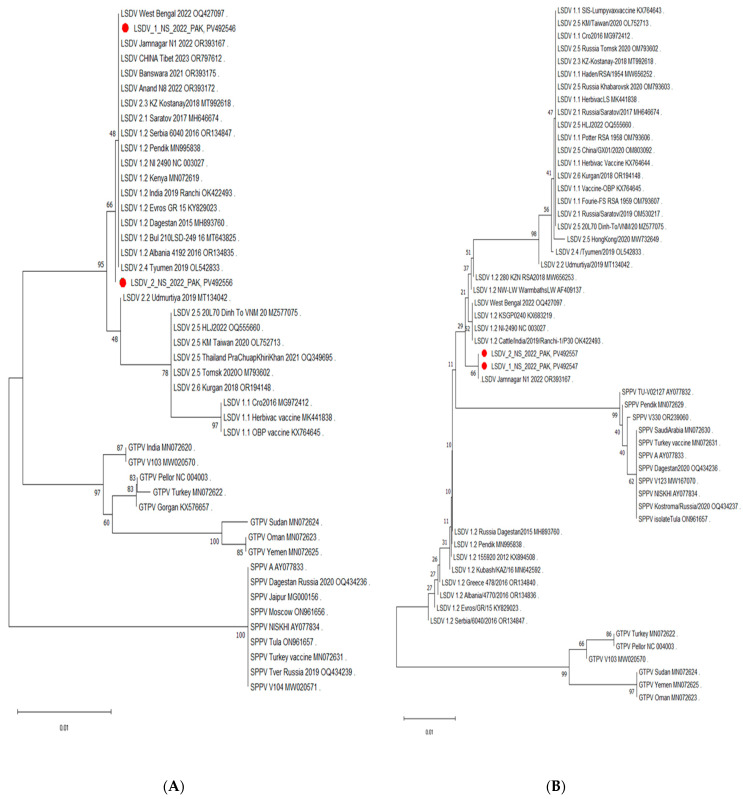
Phylogenetic tree representing (**A**) the ***B22R*** gene (**B**) the ***EEV*** gene (**C**) the ***GPCR*** gene (**D**) the ***P32*** gene and (**E**) the *RPO30* gene of isolates from Pakistan *LSDV_1_NS_2022_PAK* and *LSDV_2_NS_2022_PAK*, indicating their close relation to LSDV strains from Russia, Albania, and Serbia. The tree was generated using the neighbor-joining method; however, the phylogenetic placement of both isolates remained the same using the maximum likelihood method. The *EEV* gene made a separate sister clade with the Jamnagar isolate from India. However, the phylogenetic placement of the isolate remained the same using the maximum likelihood method as well.

**Table 1 viruses-17-01546-t001:** Primers used for phylogenetic analysis.

Serial No.	Targeted Genes	Primer Sequence (Forward)	Amplicon Size (bp) and Annealing Temperature
1	*P32*	F-5′TCGTTGGTCGCGAAATTTCAG3′ R-5′GAGCCATCCATTTTCCAACTCT3′	759, 56 °C [42]
2	*EEV*	F-5′ATGGGAATAGTATCTGTTGTATACG3′ R-5′CGAACCCCTATTTACTTGAGAA3′	930, 55 °C [43]
3	*B22R*	F-5′TCATTTTCTTCTAGTTCCGACGA3′ R-5′TTCGTTGATGATAAATAACTGGAAA3	863, 58 °C [30]
4	*RPO30*	F-5′ATTCGTTATCGCAGAACAAGG3′ R-5′CACCAACCATAGAATAGTATTGAGAC3′	1234, 55 °C [44]
5	*GPCR* *GPCR* internal sequencing primers	F5′TTAAGTAAAGCATAACTCCAACAAAAATG3′ R5′TTTTTTTATTTTTTATCCAATGCTAATACT3′ F5′GATGAGTATTGATAGATACCTAGCTGTAGTT3′ R5′TTAAGTAAAGCATAACTCCAACAAAAATG3′	1158, 50 °C [28,37]
6	*NTPase* (ORF83)	F-5′GAGAAACCGCAACAGGAAAA3′ R-5′GGATGAGCAACGAACCAACT3′	614, 60 °C [32]
7	*RPO132* (ORF116/117)	F-5′TGGAGAAATGGAAAGGGATTG3′ R-5′CAGGCGACGATGATGAAAC3′	750, 60 °C [32]
8	*VLTF-1* (ORF58/59)	F-5′TTTTATGGCGTTCCACGATT3′ R-5′CCCAACACTCTCTCGCTTCA3′	755, 60 °C [32]
9	*Finger Protein* (ORF10)	F-5′ACCCAACAACACAAGGAAGG3′ R-5′CATCGCAAACAAAGAATAAGAAAG3′	708, 60 °C [32]
10	*Ser-Thr kinase* *(ORF25/26)*	F-5′TTCGTTTTCAGCGATTTTATTT3′ R-5′AGGAGATTTTATTATGAGTGGCTT3′	739, 57 °C [45]

**Table 2 viruses-17-01546-t002:** PanCapripox real-time PCR results; *n*: number of samples tested. Data expressed as a percentage of positives.

Area of Sample Collection	Sample No. (*n*)	D5r (Cp Values) %		IC %	
		Positive	Negative	Positive	Negative
Azad Jammu and Kashmir (AJ&K)	8	75	25	87.5	12.5
Islamabad	1	100	0	100	0
Khaibar Pukhtoon Khuva (KPK)	14	35.7	64.3	42.9	57.1
Punjab	17	64.7	35.3	70.6	29.4
	40	57.5	42.5	57.5	42.5

IC: internal control value.

**Table 3 viruses-17-01546-t003:** DIVA PCR results of selected samples.

Sample Id	Location	PanCapripox			DIVA Recombinant Assay		Wolff Assay	
		D5r	Internal Control	External Control	Wild-Type and Recombinant	Neethling Vaccine	Wild-Type	Neethling Vaccine
LSDV_1_NS_2022_PAK	Punjab	31.56	36.85	29.03	29.78	Negative	31.04	Negative
LSDV_2_NS_2022_PAK	KPK *	29.79	34.68	28.45	33.18	Negative	Not Performed	Not Performed

* KPK= Khyber Pakhtunkhwa; data expressed as Cp values.

**Table 4 viruses-17-01546-t004:** Nucleotide differences between the samples from Pakistan (LSDV_1_NS_2022_PAK and LSDV_2_NS_2022_PAK) and representative sequences of the other LSDV genotypes, sheeppox and goatpox virus. Clade designations of LSDV are given in brackets.

Reference Strains and Subclades	Accession Number	Regions (Sample 1.4 (LSDV_1_NS_2022_PAK) and 3.29 (LSDV_2_NS_2022_PAK) PCR Fragment Size in Nucleotides) Number of Different Nucleotides in Isolates from Pakistan									
		*B22R* (773 *, 765 ^ǂ^)	*EEV* (877, 878)	*P32* (716, 678)	◊*RPO30* (1146)	*GPCR* (1083, 1098)	*NTPase* (575, 575)	*RPO132* (711, 729)	*VLTF-1* (735, 735)	*LAP/PHD* (692, 692)	*Ser-Thr Kinase* (657, 677)
Herbivac LS (1.1)	MK441838	11, 10	39, 39	6, 6	14	32, 33	2, 2	9, 9	4, 4	13, 13	7, 7
Cro2016 (1.1)	MG972412	11, 10	39, 39	6, 6	14	32, 33	2, 2	9, 9	4, 4	13, 13	7, 7
Neethling-LSD vaccine-OBP (1.1)	KX764645	11, 10	39, 39	6, 6	15	32, 33	2, 2	9, 9	4, 4	13, 13	7, 7
LSDV/Serbia/6040/2016 (1.2)	OR134847	0, 0	3, 3	0, 0	1	0, 0	0, 0	0, 0	0, 0	1, 1	2, 2
Kenya (1.2)	MN072619	0, 0	3, 3	0, 0	4	12, 12	1, 1	2, 2	0, 0	1, 1	1, 1
Evros GR/15 (1.2)	KY829023	0, 0	3, 3	0, 0	1	0, 0	0, 0	0, 0	0, 0	1, 1	2, 2
Saratov 2017 (2.1)	MH646674	0, 0	39, 39	5, 5	10	30, 31	2, 2	9, 9	4, 4	13, 13	7, 7
Udmurtiya/2019 (2.2)	MT134042	2, 2	37, 37	0, 0	4	32, 33	1, 1	3, 3	0, 0	13, 13	4, 4
KZ-Kostanay-2018 (2.3)	MT992618	0, 0	39, 39	1, 1	9	17, 17	2, 2	9, 9	4, 4	1, 1	7, 7
Tyumen/2019 (2.4)	OL542833	1, 1	38, 38	6, 6	8	27, 27	1, 1	9, 9	4, 4	10, 10	7, 7
CHINA/HLJ/2022 (2.5)	OQ555660	6, 6	39, 39	6, 6	13	20, 20	1, 1	2, 2	1, 1	13, 13	7, 7
Tomsk_2020 (2.5)	OM793602	6, 6	39, 39	6, 6	13	20, 20	1, 1	2, 2	1, 1	13, 13	7, 7
Kurgan 2018 (2.6)	OR194148	6, 6	39, 39	1, 1	9	12, 12	2, 2	9, 9	4, 4	1, 1	7, 7
KSGP0240 (1.2)	KX683219	1, 1	3, 3	0, 0	4	12, 12	1, 1	2, 2	0, 0	1, 1	1, 1
SPPV	MN072627	24, 23	48, 48	12, 12	61	51, 52	8, 8	36, 36	20, 20	22, 22	59, 59
GTPV	MN072622	16, 16	53, 53	7, 7	27	49, 51	8, 8	16, 16	24, 24	26, 26	34, 34

* Length of respective gene in strain *LSDV_1_NS_2022_PAK*. ^ǂ^ Length of respective gene in strain *LSDV_2_NS_2022_PAK. ◊RPO30 was amplified from only* LSDV_1_NS_2022_PAK.

## Data Availability

The data that support the findings of this study are openly available in NCBI GenBank at https://www.ncbi.nlm.nih.gov/genbank/, (accessed on 14 April 25) under the accession numbers *B22R*; PV492546, PV492556, *EEV* gene; PV492547, PV492557, *P32*; PV492548, PV492558, *RPO30*; PV492549, *GPCR*; PV492550, PV492559, *NTPase*; PV492551, PV492560, RNA polymerase 132 (*RPO132*); PV492552, PV492561, *VLTF-1*; PV492553, PV492562, *Finger Protein*; PV492554, PV492563, Ser/Thr kinase; PV492555 and PV492564 for samples LSDV_1_NS_2022_PAK and LSDV_2_NS_2022_PAK, respectively.

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
