# Peer review of "Molecular and Phylogenetic Analyses of Lumpy Skin Disease Virus (LSDV) Outbreak (2021/22) in Pakistan Indicate Involvement of a Clade 1.2 LSDV Strain"

_viruses, 2025, doi:10.3390/v17121546_

Round 1

Reviewer 1 Report

Comments and Suggestions for Authors

The study  reports on the emergence of a new lineage in Pakistan. The evidence comes from Sanger sequencing of individual genomic loci.

This was an interesting read but the way the novelty is presented leaves much to be desired. The authors fail to provide comments on the occurrence of the Middle-Eastern 1.2 lineage in Pakistan at the interface of China and India with their own genetic pools. Moreover, this idea, is poorly explained and gets lost in the describtion of general things in Discussion.

I recommend publication of the findings however the authors are encouraged to improve the following aspects:

L491 the authors claim that clade 1.2 is not surprising, which is misleading The origin of 1.2 in Pakistan  remains the principal question. Most risks come from China with 2.5 and India with KSGP due to shared long borders, whereas OR797612 from China and OQ588787 from India are already in the region.  Please include those in your analysis and discussion

L493 animal trade and cattle movement incidents are recorded across the borders - please give citations. To which scale and how it measures up with the ongoing epidemics?

Pakistan has a huge catlle and buffaloes population. The cattle is about 50 mln, the buffalo is about 37 mln.  Is there a need to import cattle from India? - the answer is evident. China - hardly. The main partners are  USA, Australia, Arab Emirates.  At the moment LSD is on the rise in Pakistan and the epidemics is far from control due to veterinary rules or the lack thereof. The authors should be carefull speculating on cattle movement without actual data.  The rewrite of the statement is required

Abstract

“clad 1.2 and shared highest homology with strains from neighboring countries, including China, India and far geographic countries, including Kenya, Russia, Serbia, Greece, Albania.” I read this sentence and understand the virus is genetically the same in general and found everywhere. The meaning of the findings is lost.  Please address the contradiction and specify clearly  to clad 1.2  (Warmbaths-like, not KSGP-like)

Is it possible to add some epidemiological data on LSD in Pakistan? Cattle and buffaloes are two target populations and they are worthwhile mentioning about.

Minor comments

 Phylogenetic trees should have a better resolution

Why not build a concatenated tree rather than multiple independent ones?

Did you see LSD in buffaloes?

Author Response

Reviewer 1 comments and rebuttal.

The study reports on the emergence of a new lineage in Pakistan. The evidence comes from Sanger sequencing of individual genomic loci.

This was an interesting read but the way the novelty is presented leaves much to be desired. The authors fail to provide comments on the occurrence of the Middle Eastern 1.2 lineage in Pakistan at the interface of China and India with their own genetic pools. Moreover, this idea is poorly explained and gets lost in the description of general things in Discussion.

I recommend publication of the findings however the authors are encouraged to improve the following aspects:

Author comments. We are thankful to the reviewer 1 for an insightful comment, we have rephrased text at various sections in paper to emphasize the emergence of clade 1.2 in Pakistan as recommended by reviewers. Please see the answers to the following comments.

Reviewer Comment: L491 the authors claim that clade 1.2 is not surprising, which is misleading the origin of 1.2 in Pakistan remains the principal question. Most risks come from China with 2.5 and India with KSGP due to shared long borders, whereas OR797612 from China and OQ588787 from India are already in the region.  Please include those in your analysis and discussion

Author Reply. The statement was not meant to undermine occurrence of clade 1.2 in Pakistan, rather to emphasize that it was not surprising to find LSDV due to cross border mobility. However as per reviewer suggestion the entire section is rephrased and presence of clade 1.2 isolates in Pakistan is discussed clearly, and two mentioned accession reports are also included in discussion and result sections under reference numbers 47 and 48. Please see  (L585- L608.).

Reviewer Comment: L493 animal trade and cattle movement incidents are recorded across the borders - please give citations. To which scale and how it measures up with the ongoing epidemics?

Pakistan has a huge cattle and buffaloes’ population. The cattle are about 50 mln, the buffalo is about 37 mln.  Is there a need to import cattle from India? - the answer is evident. China - hardly. The main partners are USA, Australia, Arab Emirates.  At the moment LSD is on the rise in Pakistan and the epidemics is far from control due to veterinary rules or the lack thereof. The authors should be careful speculating on cattle movement without actual data.  The rewrite of the statement is required

Author Reply: Entire section is rephrased as per reviewer suggestions and improved to include recommendation please have a look at L 585- 608, in discussion section.

Reviewer Comment:

Abstract

“clad 1.2 and shared highest homology with strains from neighbouring countries, including China, India and far geographic countries, including Kenya, Russia, Serbia, Greece, Albania.” I read this sentence and understand the virus is genetically the same in general and found everywhere. The meaning of the findings is lost.  Please address the contradiction and specify clearly to clad 1.2 (Warmbaths-like, not KSGP-like)

Author Reply. The abstract is rephrased as per reviewer 1 and other reviewers’ suggestions. The findings are highlighted, and results are phrased appropriately. Please see lines L20-L35

Reviewer Comment:

Is it possible to add some epidemiological data on LSD in Pakistan? Cattle and buffaloes are two target populations, and they are worthwhile mentioning about.

Author Reply.

Pakistan outbreak epidemiological data is added to the introduction section L42-L52. Please have a look. The data is updated regrading prevalence in buffaloes (L50-52).

Reviewer 1 Minor Comments:

Reviewer Comment:  Phylogenetic trees should have a better resolution

Author Reply. The resolution of phylogenetic trees is improved please see Figure 3A-E pages 16-18, revised manuscript all changes accepted.

Reviewer Comment: Why not build a concatenated tree rather than multiple independent ones?

Author Reply. We do agree with the other that concatenation is a powerful tool to use in this situation. However, this was not done due to the unknown evolutionary speed  of the different genes which could influence the correctness of the overall tree. Furthermore, the individual trees facilitate comparison with other studies using one or more of  these genes.

Reviewer Comment: Did you see LSD in buffaloes?

Author Reply. We collected samples randomly from different areas in Punjab KPK, Sindh and AJ and K. Since this research was conducted in absence of any funding, areas were selected based on ease of accessibility. We did not observe any infected buffalo populations in those areas as the overall prevalence of LSDV in buffalo is low with only one report from Pakistan (9.6%, https://doi.org/10.1007/s11250-025-04557-7). It was acceptable as our sampling was not intended to comprehend prevalence of LSDV infection in different areas and species. This information is added in Introduction L50-52, Reference 5 in manuscript.

Reviewer 2 Report

Comments and Suggestions for Authors

In this manuscript, Ferdoos et al. describe the molecular and phylogenetic analyses of a lumpy skin disease outbreak in Pakistan.

They found that 57% of collected samples were positive in an pan-capripox qPCR assay. They went on to PCR amplify and sequence 10 genomic regions of two isolates. Sequence and phylogenetic analysis of these fragments revealed that the sequences belonged to LSDV clade 1.2. This main conclusion is supported by the data.

In addition, the authors compared the putative amino acid changes in the encoded sequence fragments and speculate about their biological impact. The description of these amino acid changes is fine, but the speculation about their potential impact is wildly speculative. Examples in the result section include “Since both aminoacids have hydropilic nature and are polar such a substitution may not have great impact (Figure 2C).” or, “Such a substitution can alter protein behaviour and antigenic properties as both aminoacids have opposite charges.” Or” however another Indian isolate and a chinese isolate had a substituion at position 30 (Y30H). Sucha a substitution may reflect changes in electronic properties , binding affinity and stability/folding of protein, and hence can be of evolutionary significans. (Supplementry figure 3B).” , These wild speculations should not be part of the result section. Even in the discussion section these speculations seems misplaced because the authors just state platitudes such as “The nucleotide changes in the EEV fragment, encoding parts of LD126 (EEV) and LD127 proteins, of the Pakistan samples resulted in only one amino acid change, namely a lysine to glutamic acid. This is an interesting change, as lysine is negatively  charged while glutamic acid is positively charged.” If the authors wanted to speculate about a certain effect of a substitution, they should back up their assertion with better reasoning.

In the abstract, the authors write:” Pakistan strains belonged to clad 1.2 and shared highest homology with strains from neighboring countries, …”. “highest homology” should be changed to something like “highest sequence identity”. There are not different degrees of homology. Sequences are either homologous or they are not.

All phylogenetic analyses must contain bootstrap values! Some analyses in the supporting information are missing this information. Please also include statements on how the trees were rooted.

Comments on the Quality of English Language

The English has to be substantially revised. The manuscript is full of errors. It seems that not even a spellcheck was performed. In the quotes above, examples of these errors are included.

Countries and locations are written in capital letters, e.g “china and and vietnam”, it is lumpy skin disease virus and not Lumpy skin disease virus. It is “sheeppox virus”, not Sheeppox virus”.

Author Response

Reviewer 2 comments and rebuttal.

General Comments: In this manuscript, Ferdoos et al. describe the molecular and phylogenetic analyses of a lumpy skin disease outbreak in Pakistan. They found that 57% of samples collected were positive in a pan-capripox qPCR assay. They went on to PCR amplify and sequence 10 genomic regions of two isolates. Sequence and phylogenetic analysis of these fragments revealed that the sequences belonged to LSDV clade 1.2. This main conclusion is supported by the data.

Reviewer comment 1; In addition, the authors compared the putative amino acid changes in the encoded sequence fragments and speculate about their biological impact. The description of these amino acid changes is fine, but the speculation about their potential impact is wildly speculative.

Author Reply: Thank you for pointing out this issue. The speculations/ deductions were made after studying similar amino acid changes in published literature. However, mistakenly the references were not added. We have added appropriate reference at each deduction/ speculation we made from published literature.

Reviewer: Examples in the result section include “Since both amino acids have hydrophilic nature and are polar such a substitution may not have great impact (Figure 2C).”

Author reply: The statement has been corrected and revised based on published literature. Please see reference 56 in manuscript, L336-340.

(T.-J. Zhao, Y. Liu, Z. Chen, Y.-B. Yan, and H.-M. Zhou, “The evolution from asparagine or threonine to cysteine in position 146  contributes to generation of a more efficient and stable form of muscle creatine kinase in higher vertebrates.,” Int J Biochem Cell Biol, vol. 38, no. 9, pp. 1614–1623, 2006, doi: 10.1016/j.biocel.2006.04.002 )

Reviewer: “Such a substitution can alter protein behaviour and antigenic properties as both amino acids have opposite charges.” Line 254-55 original manuscript.

Author Reply. This statement was drawn after studying following publications. Such a mutation is reported to alter protein properties. Statement has been changed, and references are added please see references 52, 53 and 54. in revised manuscript L 320-329.

52] D. S. Meyer et al., “Expression of PIK3CA mutant E545K in the mammary gland induces heterogeneous tumors but is less potent than mutant H1047R,” Oncogenesis, vol. 2, no. 9, pp. e74–e74, 2013, doi: 10.1038/oncsis.2013.38.

53       C.-H. Huang, D. Mandelker, S. B. Gabelli, and L. M. Amzel, “Insights into the oncogenic effects of /PIK3CA/ mutations from the structure of p110α/p85&alpha,” Cell Cycle, vol. 7, no. 9, pp. 1151–1156, May 2008, doi: 10.4161/cc.7.9.5817.

[54]     J. D. Carson et al., “Effects of oncogenic p110α subunit mutations on the lipid kinase activity of phosphoinositide 3-kinase,” Biochemical Journal, vol. 409, no. 2, pp. 519–524, 2008.

Reviewer: Or” however another Indian isolate and a Chinese isolate had a substitution at position 30 (Y30H). Such substitution may reflect changes in electronic properties, binding affinity and stability/folding of protein, and hence can be of evolutionary significance. (Supplementary figure 3B).

Author reply: This statement was based on following publication. Please see reference 57 in manuscript. Statements have been refined, references are added and speculations are changed to possibilities. Please see lines 344-351.

Reference 54 in revised manuscript

  1. J. Betts and R. B. Russell, “Amino Acid Properties and Consequences of Substitutions,” in Bioinformatics for Geneticists, Wiley, 2003, pp. 289–316. doi: 10.1002/0470867302.ch14.

These wild speculations should not be part of the result section. Even in the discussion section these speculations seem misplaced because the authors just state platitudes such as

“The nucleotide changes in the EEV fragment, encoding parts of LD126 (EEV) and LD127 proteins, of the Pakistan samples resulted in only one amino acid change, namely a lysine to glutamic acid. This is an interesting change, as lysine is negatively charged while glutamic acid is positively charged.” If the authors wanted to speculate about a certain effect of a substitution, they should back up their assertion with better reasoning.

Author reply: As mentioned earlier these speculations/ deductions were made based on published literature where similar substitutions are known to affect protein properties. References have been added at this place as well in discussion section.

Reviewers’ comments: In the abstract, the authors write:” Pakistan strains belonged to clad 1.2 and shared highest homology with strains from neighbouring countries, …”. “Highest homology” should be changed to something like “highest sequence identity”. There are not different degrees of homology. Sequences are either homologous or they are not.

Author Reply: The entire Abstract is revised as per reviewers’ suggestions. These statements have been re-phrased to make a better impact please see lines L 20-L35 in revised manuscript.

Reviewer comment: All phylogenetic analyses must contain bootstrap values! Some analyses in the supporting information are missing this information. Please also include statements on how the trees were rooted.

 Author Reply: The information in supplementary file is updated. All phylogenetic trees have boots trap values now. Please see supplementary material Figure 4A-E. All trees were unrooted.

Comments on the Quality of English Language

Reviewer comment The English must be substantially revised. The manuscript is full of errors. It seems that not even a spellcheck was performed. In the quotes above, examples of these errors are included.

Author Reply: The entire manuscript is checked thoroughly for grammatical and spelling errors; we have improved language to the best of our knowledge. Please let us know if further improvement is required.

Reviewer comment: Countries and locations are written in capital letters, e.g “china and and vietnam”, it is lumpy skin disease virus and not Lumpy skin disease virus. It is “sheeppox virus”, not Sheeppox virus”.

Author Reply: We have removed the capitalization, grammatical and typing errors to best of our knowledge.

Reviewer 3 Report

Comments and Suggestions for Authors

The article "Molecular and Phylogenetic Analyses of Lumpy Skin Disease Virus (LSDV) Outbreak (2021/22) in Pakistan Indicates Involvement of Clade 1.2 LSDV Strain" written by Ferdoos et al. conducted study in three major provinces of Pakistan to detection and phylogenetically classify LSDV strains based on 10 genomic sequences. The article needs minor revision before publication of this manuscript. The authors can find the comments in the attached PDF for further improvement of their manuscript. 

Author Response

Reviewer 3; The authors can find the comments in the attached PDF for further improvement of their manuscript. 

Author Reply: Following comments have been copied from the attached PDF.

Comments from PDF

1) The abstract contains too much sentence on background and methods but very few contents of results and lacks concluding statement. It needs to be updated,

Author Reply: As per reviewers comments the abstract has been rephrased to include more results and less background. Few changes have been made as per reviewer 1 suggestions. Please see L20-L35 in revised manuscript file.

2) who vast majority?

Author Reply: The sentence is rephrased please see L 20-22 and L38 in abstract.

3) Why is the whole text italic? only gene name should be italic.

Author Reply: corrected, please see L85-90.

4) There should be no space like this in the whole manuscript.

Author Reply: Corrected throughout manuscript. Please see L89

5) place comma here.

Author Reply: Corrected please see L93. Statement rephrased.

6) Figure quality is not good. Also mention the number of samples collected from each site.

Author Reply: The figure is changed, and a better-quality map is added. Also, the number of samples collected from each location is mentioned in description and on map. Please see page 3, figure 1, L101-105.

7) Is this being total samples? What is the statistical significance of this sample size and which strategy was used to calculate the samples being collected?

Author Reply: Sample collection, type, number and collection sites have been updated. Please see lines 101-105 and L121-127. The Samples were randomly collected based on severity of disease in that area.

8) What you mean by specialized? What's special in this PCR?

Author Reply: The term specialized was coined to this real-time qPCR in lieu of its specificity to differentiate between pox viruses up to genus Level. This probe-based qPCR was fist time developed and validated by Wolff in 2021 (Reference 42, revised manuscript). It was fist of its kind at that time.

Similarly, DIVA PCR was developed and validated by Andy Haegeman in 2023 which was able to differentiate between different species of LSDV genus using probes directed to differential sites in different species. These features make it specialized for LSDV genus and species identification. Please see references 41,42,43, revised manuscript.

9) i think it's inappropriate to write like this. it can be written as author name et al.

Author Reply: Corrected; reference and authors are mentioned, please see L134-141 revised manuscript.

10) I think it should be "Molecular detection and sequence analysis of 10 genes".

Author Reply: The statement is rephrased as per reviewer suggestion. Please see L142

11) regions or genes?

Author Reply: Few genes were not complete; therefore, we used term region, however, it is rephrased. Please see L 143, section 2.4.1.

12) Short description should be written here as well.

Author Reply: Description is added. Please see L146-L152, section 2.4.1.

13) What you mean by this gene?

Author Reply: GPCR is a long fragment, we know from the past that it is sometimes difficult to have the complete sequence for this region. To solve this, we use internal sequencing primers. This is why you do not see an amplicon light for these primers

14) why only one sample? There was outbreak and you just collected one sample?

Author Reply: This table represents number of samples tested through real time PCR, not the total number of samples collected. Moreover, when we collected samples, vaccination was in progress in majority farms in ICT therefore only 2 samples were included in analysis.

15) why this text is below the table. is it footnote or continuation of above text? It should be above the table, not below.

Author Reply: The text is moved above the table please check L239-240, revised manuscript

16) why some words have capital first alphabets while other has small?

Author Reply: Corrected. Please see L246

17) What is Cp-value? Did you explain elsewhere?

Author Reply: Cp value is the crossing point where fluorescence of sample rises above background levels. This information is updated in L235-236, section 3.1.

18) Why you selected too few samples and what's the reason of selecting these two samples?

Author Reply: Why too few samples? Out of all positive samples majority were swabs, and there was little material recovered, leaving less choice for further analysis.

The criterion of selection is mentioned in the same two lines please see lines L211-212. The criterion was good Cp value and availability of enough material for further analysis.

19) It's inappropriate to write like this.

Author Reply: It is rephrased please have a look L214.

20) The conclusion should include objective, main findings, future strategies, and recommendations.

Author Reply: Conclusion is rephrased please see lines L648-656.

Reviewer 4 Report

Comments and Suggestions for Authors

The manuscript entitled “Molecular and Phylogenetic Analyses of Lumpy Skin Disease Virus (LSDV) Outbreak (2021/22) in Pakistan Indicates Involvement of Clade 1.2 LSDV Strain”, the authors describe the clustering of two isolates obtained from outbreaks in Pakistan based on ten gene or partial gene regions. The findings are interesting and could benefit our current understanding of the molecular epidemiology of LSDV, but the manuscript requires significant editing. Primarily, the manuscript requires significant language editing and curation. Additionally, the figures are not legible and should be edited.

Comments and suggestions.

  1. Line 50. The 2.4 Kb inverted terminal regions do consist of coding regions, thus they do not flank the coding region.
  2. Lines 59 - 62. Requires a refence for cluster 1.1 and the references for Cluster 1.2 is not correct. Neither are all the six sub-clades of cluster 2 referenced.
  3. Figure 1 is not legible. Please change the background or enhance / contrast the writing.
  4. Table 1. What was the “6. Sequencing” primers used for? Please clarify.
  5. Lines 175-178. What is the cut-off for the RT-PCR? Ct values at 35.8 +4 and 38.67+4 indicate values above 40 (the usual cut-off for positive samples).
  6. Lines 188 – 190. This sentence should be re-written to include “classified samples either as”
  7. Refrain from referring to samples as a nationality (for example Pakistani sample), but rather an sample from Pakistan.
  8. Line 267: Reference required.
  9. Figures 2 and 3 in relation to table 4. It would be easier to compare the sequences if the different gene regions of the same isolates were compared each time. Table 4 uses 14 LSDV, one SPPV and one GTPV to compare against the new sequences from Pakistan. These sequences with the same name and Genbank accession number, could be used in the subsequent figures, rather than the multiple partial sequences with short descriptions such as on the country name. This is especially important, since multiple LSDVs belonging to different clusters have been described in certain countries (for example Russia).
  10. Figure 2A, B and C. Could the query sequence (sample from Pakistan) be on the top of each alignment for comparison?
  11. Lines 334-341. The sentence “Examples of the evolutionary historical inference are depicted in Figure 1, 2, and 3” need to be revised since this makes no sense. Figure 1 is a map, figure 2 contains alignments and figure 3 phylogenetic trees.
  12. Lines 336-341. The concept of different sequences that cluster with the newly described samples have been addressed in comment no10. It would make more sense if the different gene regions of the same isolates (representing the different clusters) are used during the phylogenetic analyses. Similarly, as previously mentioned in comment no 10, certain countries have reported the circulation of multiple strains of LSDV belonging to different clusters. Thus referring to sequence obtained from Russia or Kenya could be misleading since multiple LSDV clusters have been described to circulate there. It would make more sense to refer to the specific isolate (for that country) that the new samples share a close relationship to.
  13. Figure 3. The images are not clear and should be revised and enhanced. Perhaps by using less sequences or collapsing the SPPV and GTPV groups. Change the name of the new sequences obtained from the samples in Pakistan to include the GenBank accessions, similar to the sequences obtained from Genbank. Please also remove the “gene name” and “con” at the end of the names of the newly described sequences. Additionally, cluster names in brackets will be useful.
  14. Line 476: Gene names are usually Italic. Could you please include the Open Reading frame number for the different gene regions used?
  15. Lines 502-505. This is not correct. Lysine is positive and glutamic acid is negative.
  16. Lines 503 and 508. What is frag4 and frag 5?
  17. Supplementary figures 3 have the same comments as 11. Supl Fig 3C, 3D and 3E describes amino acid changes in yellow, but nothing is highlighted or different. Please correct.
  18. Supplementary figures 3 have the same comments as 14. Please clarify the sequences referring to “inhouse” and “con”?
Comments on the Quality of English Language
  1. The manuscript needs language revision and editing. Certain examples are listed below:

Line 34; clad – clade.

Line 35: Far geographic countries –

Line 49: LSDV carries 151kb genome –

Line 57: Southeast Asia and South Asian countries – South and Southeast Asian countries

Lines 109 and 118: Elsewhere – previously

Line 121: prob – probe

Lines 122-123: Endemic – LSDV is not endemic to these countries, rather use observed or circulating.

Line 239: siolates – isolates.

Line 242 - 259: aminoacid – amino acids

Line 243: chnages – changes

Lines 246 – 272: Country names should be with a capital letter: China, India, Pakistan, Vietnam

Line 253: Indiana nd Pakistani – India and Pakistan.

Line 271: sucha – such a

Line 340: kennyan

Line 472: “]. and”

Author Response

Reviewer 4 comments and rebuttal.

Comments and Suggestions for Authors

General Comments

The manuscript entitled “Molecular and Phylogenetic Analyses of Lumpy Skin Disease Virus (LSDV) Outbreak (2021/22) in Pakistan Indicates Involvement of Clade 1.2 LSDV Strain”, the authors describe the clustering of two isolates obtained from outbreaks in Pakistan based on ten gene or partial gene regions. The findings are interesting and could benefit our current understanding of the molecular epidemiology of LSDV, but the manuscript requires significant editing. Primarily, the manuscript requires significant language editing and curation. Additionally, the figures are not legible and should be edited.

Author Reply: We are grateful for these insightful comments by the reviewer, we have improved the resolution of figures in the manuscript and revised English language throughout manuscript. The language is improved, and grammatical and spelling errors are corrected to the best of our knowledge. 

Comments and suggestions.

  1. Line 50. The 2.4 Kb inverted terminal regions do consist of coding regions, thus they do not flank the coding region.

Author Reply: statement is rephrased and corrected please se L56-57, revised manuscript file.

  1. Lines 59 - 62. Requires a refence for cluster 1.1 and the references for Cluster 1.2 is not correct. Neither are all the six sub-clades of cluster 2 referenced.

Author Reply: References have been added where recommended. Please see reference 18-26

  1. Figure 1 is not legible. Please change the background or enhance / contrast the writing.

Author Reply: Figure 1 is changed, sampling sites and number have been added, please see Figure 1 Page 4.

  1. Table 1. What was the “6. Sequencing” primers used for? Please clarify.

Author Reply: Corrected, please table 1. These are internal sequencing primers for GPCR gene.

  1. Lines 175-178. What is the cut-off for the RT-PCR? Ct values at 35.8 +4 and 38.67+4 indicate values above 40 (the usual cut-off for positive samples).

Author Reply: The D5R PCR is a part of an RT-PCR panel consisting of three real-time PCRs. The samples are first tested with the D5R PCR. Samples with a Cp value of < 37 are considered positive; Samples with a Cp > 37 are considered doubtful. These doubtful samples are then tested with both other real-time PCRs. If at least one of these 2 real-time PCRs have a Cp value, it is considered to be positive.

  1. Lines 188 – 190. This sentence should be re-written to include “classified samples either as”

Author Reply: Rephrased please see lines 251-253.

  1. Refrain from referring to samples as a nationality (for example Pakistani sample), but rather a sample from Pakistan.

Author Reply: Corrected throughout manuscript except where necessary.

  1. Line 267: Reference required.

Author Reply: References are updated for entire section please see reference number 57 for this line  

57) G. Kok et al., “Isoleucine-to-valine substitutions support cellular physiology during isoleucine deprivation,” Nucleic Acids Res, vol. 53, no. 1, p. gkae1184, Jan. 2025, doi: 10.1093/nar/gkae1184.

  1. Figures 2 and 3 in relation to table 4. It would be easier to compare the sequences if the different gene regions of the same isolates were compared each time. Table 4 uses 14 LSDV, one SPPV and one GTPV to compare against the new sequences from Pakistan. These sequences with the same name and Genbank accession number, could be used in the subsequent figures, rather than the multiple partial sequences with short descriptions such as on the country name. This is especially important, since multiple LSDVs belonging to different clusters have been described in certain countries (for example Russia).

Author Reply: table 4 represents the details of one-member of each clade and sub clade, that is compared to isolates from Pakistan. This table gives an overall comparison of similarities and differences among various LSDV clades, in the stated 10 genes. It comprehends iter clade similarities and differences.

However, the protein comparison of isolates from Pakistan is carried out between members of same clade (1.2) that are closely related to these isolates. This comparison gives an information related to divergence of isolates from Pakistan from its close relatives in neighboring countries.

As per reviewers’ recommendations the complete names and accessions of isolates are included in Figure 2 A, B and C. Please see page 12-13 of revised manuscript.

  1. Figure 2A, B and C. Could the query sequence (sample from Pakistan) be on the top of each alignment for comparison? Table 4 represents each meme

Author Reply: The software automatically positions accessions in the alignment according to the algorithm used, therefore we have highlighted the accession of isolate from Pakistan so that viewers can easily understand. Please see figure 2 A, B and C at page 12-13 of revised manuscript.

  1. Lines 334-341. The sentence “Examples of the evolutionary historical inference are depicted in Figure 1, 2, and 3” need to be revised since this makes no sense. Figure 1 is a map, figure 2 contains alignments and figure 3 phylogenetic trees.

Author Reply: Figure number updated. Figure 3 A-E. Please see L357, revised manuscript.

  1. A) Lines 336-341. The concept of different sequences that cluster with the newly described samples have been addressed in comment no10. It would make more sense if the different gene regions of the same isolates (representing the different clusters) are used during the phylogenetic analyses.

Author Reply: Different regions of both isolates are included in analysis. The phylogenetic trees have included same gene from all clusters in each map. For example, EEV gene phylogenetic tree at page 15, figure 3B included clade number along with isolate accessions. Each tree included members of different clades. However, Many of the sequence information included in the tree is from partial sequencing and therefore the sequence for all the different regions is not always available in the public database

  1. B) Similarly, as previously mentioned in comment no 10, certain countries have reported the circulation of multiple strains of LSDV belonging to different clusters. Thus, referring to sequence obtained from Russia or Kenya could be misleading since multiple LSDV clusters have been described to circulate there. It would make more sense to refer to the specific isolate (for that country) that the new samples share a close relationship .

Author Reply: As per reviewers’ suggestion the isolates are named according to NCBI, and their accession numbers have been added rather than country name. Please see L288-308, revised manuscript. Same is revised throughout manuscript.

  1. Figure 3. The images are not clear and should be revised and enhanced. Perhaps by using less sequences or collapsing the SPPV and GTPV groups. Change the name of the new sequences obtained from the samples in Pakistan to include the GenBank accessions, similar to the sequences obtained from Genbank. Please also remove the “gene name” and “con” at the end of the names of the newly described sequences. Additionally, cluster names in brackets will be useful.

Author Reply: Resolution of images is improved and GenBank accessions are added as advised. Since gene sequences of isolates from Pakistan matches with isolates of different clades therefore clade and sub clades are mentioned with isolate accession numbers in trees.

  1. Line 476: Gene names are usually Italic. Could you please include the Open Reading frame number for the different gene regions used?

Author Reply: Names are italicized, ORF numbers are mentioned and reference is provided. Please see L529-533.

  1. Lines 502-505. This is not correct. Lysine is positive and glutamic acid is negative.

Author Reply: Corrected please see L575.

  1. Lines 503 and 508. What is frag4 and frag 5?

Author Reply: Corrected; Fragment 4 refers to finger protein (ORF10) and Frag 5 refers to serine /threonine kinase (ORF25). Information is updated in L574 and L579.

  1. Supplementary figures 3 have the same comments as 11. Supl Fig 3C, 3D and 3E describes amino acid changes in yellow, but nothing is highlighted or different. Please correct.

Author Reply: The software automatically positions accessions in the alignment according to the algorithm used; therefore, we have highlighted the accession of isolate from Pakistan so that viewers can easily understand. Please see figure 3 A- E at page 12-13 of revised manuscript. When no highlight is present it reflects no difference in sequence.

  1. Supplementary figures 3 have the same comments as 14. Please clarify the sequences referring to “inhouse” and “con”?

Author Reply: I think reviewer meant Figure 4 in this comment. We have included all the recommendations of the reviewer. Names and accessions have been updated; resolution of figures is improved. Please see figure 4 A-E in supplementary file.

Comments on the Quality of English Language

  1. The manuscript needs language revision and editing. Certain examples are listed below:

Author Reply: The language of manuscript is thoroughly revised. We tried to remove all the spelling and typing mistakes as well as grammatical errors throughout manuscript.

Line 34; clad – clade.

Author Reply: Corrected, the Abstract is re written as per other reviewers’ suggestions please see L 20-34.

Line 35: Far geographic countries –

Author Reply: Corrected; Abstract is re written

Line 49: LSDV carries 151kb genome –

Author Reply: Corrected

Line 57: Southeast Asia and South Asian countries – South and Southeast Asian countries

Author Reply: Corrected

Lines 109 and 118: Elsewhere – previously

Author Reply: Corrected L 127

Line 121: prob – probe

Author Reply: Corrected L140

Lines 122-123: Endemic – LSDV is not endemic to these countries, rather use observed or circulating.

Author Reply: Corrected; 141

Line 239: siolates – isolates.

Author Reply: Corrected L236

Line 242 - 259: aminoacid – amino acids

Author Reply: corrected

Line 243: chnages – changes

Author Reply: corrected L265

Lines 246 – 272: Country names should be with a capital letter: China, India, Pakistan, Vietnam

Author Reply: corrected throughout manuscript

Line 253: Indiana nd Pakistani – India and Pakistan.

Author Reply: Corrected

Line 271: sucha – such a

Author Reply: Corrected

Line 340: kennyan

Author Reply: Corrected L399, this section is rephrased

Line 472: “]. and”

Author Reply:  Corrected

Round 2

Reviewer 1 Report

Comments and Suggestions for Authors

The authors have addressed the raised concerns and  is now publication-ready.

Author Response

Related comments and responses are available in the attached PDF.

Reviewer 2 Report

Comments and Suggestions for Authors

In their revision, Ferdoos et al. provide references for the speculations about the potential impact of putative amino acid variations. The major problem is that the references are generic in nature and are not specific for the specific poxvirus genes, that is to say, they cite papers of non-poxvirus proteins where a certain amino acid variation had an effect on protein function. Just because an amino acid change is non-conservative does not mean that this variation is potentially more consequential. There are other examples in the literature where more conservative amino acid variations also affected protein function, for example, isoleucine to valine, valine to isoleucine, or valine to alanine affected the function of a poxvirus host range protein (PMID: 33444388). This work does not need to be cited but just illustrates that statements such the following should be avoided “Another amino acid substitution, I168M, is present in the the isolate from Pakistan 328 (LSDV_2_NS_2022_PAK, PV492557), India (LSDV/2022/Jamnagar/N1,OR393167) and 329 Kenya (Kubash/KAZ/16, MN642592). … This change has probably less implications, as illustrated by Ohmura (2001) mainly because both amino acids are hydrophobic and neutral in charge.”

I maintain that these speculations are completely out of place in the results section. Even in the discussion section, these speculations are inappropriate, unless the speculations directly pertain the viral proteins. Just because another study of a totally unrelated protein found that an amino acid change affected the function of that protein, inferring that it has a similar effect on an unrelated LSDV protein is preposterous. I do not see the need to add such wild speculations to the manuscript, and the manuscript is stronger without those.

Author Response

(The authors gave the same response as above.)

Reviewer 4 Report

Comments and Suggestions for Authors

The manuscript entitled “Molecular and Phylogenetic Analyses of Lumpy Skin Disease Virus (LSDV) Outbreak (2021/22) in Pakistan Indicates Involvement of Clade 1.2 LSDV Strain”, the authors describe the clustering of two isolates obtained from outbreaks in Pakistan based on ten gene or partial gene regions. The manuscript has received significant language editing and curation, and the figures have been improved, but the following concerns should be addressed prior to recommending the manuscript for publication.

  1. Samples do not have a nationality (for example Pakistani, Indian or Russian sample). This should be changed to samples from Pakistan, India or Russia.
  2. Line 30: Change “2021/11” to “2021/22”
  3. Lines 45 to 47. Be consistent with the punctuation: “74,590”, ”6,351” and “7,500”.
  4. Lines 50 to 52. Require additional explanation of the statement. What is the proportion of cattle to water buffaloes?
  5. Lines 102 to 104. Rather change the sample numbers to (n = X).
  6. Line 148. Remove reaction after PCR, since this is redundant.
  7. Line 151. “appropriate denaturation” should probably be “appropriate annealing temperature”. Perhaps include this temperature in the table for each primer set.
  8. Table 2. Include % in the headings.
  9. Lines 559 to 561. Include the Genebank accessions of P32 sequences from Pakistan in the analyses (OQ566164, OQ566165, OQ589501, OQ589502, OP807845 - OP807849) (Jabbar et al 2025).
  10. Supplementary figures 3. Please clarify the sequences referring to “inhouse” and “con”?

Author Response

(The authors gave the same response as above.)
